# Critical Role of Cathepsin L/V in Regulating Endothelial Cell Senescence

**DOI:** 10.3390/biology12010042

**Published:** 2022-12-26

**Authors:** Chan Li, Zhaoya Liu, Mengshi Chen, Liyang Zhang, Ruizheng Shi, Hua Zhong

**Affiliations:** 1Department of Cardiovascular Medicine, Xiangya Hospital, Central South University, No. 87 Xiangya Road, Changsha 410008, China; 2National Clinical Research Center for Geriatric Disorders, Xiangya Hospital, Changsha 410008, China; 3Department of Geriatrics, Third Xiangya Hospital, Central South University, No. 138 Tongzipuo Road, Changsha 410013, China; 4Department of Epidemiology and Health Statistics, Xiangya School of Public Health, Central South University, Changsha 410028, China; 5Hunan Provincial Key Laboratory of Clinical Epidemiology, Central South University, No. 138 Tongzipuo Road, Changsha 410013, China; 6Department of Cardiovascular Medicine, Third Xiangya Hospital, Central South University, No. 138 Tongzipuo Road, Changsha 410013, China

**Keywords:** CTSV, cellular senescence, aging, ALDH1A2, retinoic acid

## Abstract

**Simple Summary:**

Endothelial cell senescence has been considered as an initiation in the progress of vascular aging leading to the advancement of cardiovascular diseases, while the mechanism of endothelial cell senescence remains elusive. This study aimed to investigate the critical role of cathepsinL/V in endothelial cell senescence. We found that cathepsinL/V was decreased in senescent endothelial cells, which enhanced aldehyde dehydrogenase 1 family member A2 (ALDH1A2) expression and activated AKT/ERK1/2-P21 pathway, and therefore promoted cellular senescence, which may play an important role in vascular aging. These findings suggest that cathepsinL/V may be a potential therapeutic target in endothelial cell senescence.

**Abstract:**

The senescence of vascular endothelial cells (ECs) is characterized as a hallmark of vascular aging, which leads to the initiation, progress, and advancement of cardiovascular diseases. However, the mechanism of the ECs senescence remains elusive. In this study, thoracic aortas were separated from young (8-week-old) and aged (18-month-old) mice. Decreased Ctsl expression and increased vascular remodeling were observed in senescent aorta. H_2_O_2_ was used to induce human umbilical vein endothelial cells (HUVECs) senescence, as shown by increased SA-β-gal positive cells and upregulated p21 level. CTSV significantly decreased after H_2_O_2_ treatment, while over-expression of CTSV by adenovirus reduced cellular senescence. RNA sequencing analysis was conducted subsequently, and ALDH1A2 was observed to significantly increased in H_2_O_2_ group and decreased after over-expression of CTSV. This result was further confirmed by RT-PCR and WB. Moreover, over-expression of CTSV reduced the increase of ERK1/2 and AKT phosphorylation induced by H_2_O_2_. Additionally, retinoic acid (RA), the major production of ALDH1A2, was added to CTSV over-expressed senescent HUVECs. Administration of RA activated AKT and ERK1/2, induced the expression of p21, and enhanced SA-β-gal positive cells, while not affecting the expression of CTSV and ALDH1A2. These results were further confirmed in doxorubicin (DOX)-induced senescent ECs. In conclude, we have identified that Ctsl/CTSV plays a key role in ECs senescence by regulating ALDH1A2 to activate AKT/ ERK1/2-P21 pathway. Therefore, targeting Ctsl/CTSV may be a potential therapeutic strategy in EC senescence.

## 1. Introduction

With the advancement of the medicine and public health, the human life expectancy is gradually increasing in the world and aging has been recognized as a global phenomenon over the past decades [1]. According to the report of the World Health Organization, the world’s population of people aged 80 years or even older is predicted to triple between 2020 and 2050 to reach 426 million [2]. This demographic shift contributes to major challenges and economic burdens to society. It has been demonstrated that aging is the single most important risk factor in morbidity and mortality of cardiovascular disease (CVD), with the risk approximately tripling with each decade of life [3]. CVD, the leading cause of death worldwide, is projected to be responsible for more than 23 million deaths in 2030 around the world [4].

Vascular aging, a key threshold of overall aging, plays a crucial role in the development and deterioration of CVD [5]. Aging promotes functional and structural alterations of the vasculature including impairment endothelium function and angiogenesis, increased luminal diameter, vascular stiffening, and extracellular matrix (ECM) reorganization [6]. Especially, the senescence of vascular endothelial cells (ECs) has been considered as an initiation in the progress of vascular aging leading to the advancement of CVD [7]. ECs senescence is characterized as becoming flatter and polypoid nucleus, accompanied by regulation in angiogenesis, proliferation, migration, and cytoskeleton integrity [8]. Recent studies reported that oxidative stress, activated rein angiotensin aldosterone system (RASS), and calcium signaling contribute to the progress of senescence [9], while the mechanism of ECs senescence is not fully elucidated.

Cellular senescence refers to a form of stable cell cycle arrest [10]. It may occur after repeated cellular division followed by progressive attrition of telomeres and activation of DNA damage response pathways, which is known as replicative senescence. Furthermore, it may also result from cellular stress such as radiation, oxidative stress, or cytotoxic drugs, which is known as stress-related senescence. These two methods may activate p21 by inducing oxidative stress, DNA damage, or telomere dysfunction, and eventually result in cell cycle arrest [11]. Many in vitro models have been used to induce cellular senescence via these mechanisms. For instance, hydrogen peroxide (H_2_O_2_) is used to induce senescence by triggering oxidative stress, while radiation is often used to trigger DNA damage. In addition, doxorubicin (DOX) can promote DNA damage, oxidative stress, and telomere dysfunction at the same time [12]. Since senescent cells cannot be defined by a single trait, multiple hallmarks should be tested to confirm the senescent phenotype. A guide to accessing cellular senescence suggests the verification of at least three different traits, including cell cycle arrest, structural change, and additional trait. Increased expression of p53, p16, and p21 are the main markers of cell cycle arrest, while the increase of SA-β-galactosidase is one of the most widely used markers, which reflects increased lysosomal mass in senescent cells. As for additional traits, senescent cells may produce a complex secretome, such as IL-1β, IL-6, and ICAM-1, known as senescence-associated secretory phenotype (SASP), which is widely used as a secondary marker to access cellular senescence [13].

The cysteine cathepsin family, localized in lysosomes and released or expressed by cells, serves to degrade intracellular proteins and further regulate ECM remodeling [14]. It has been shown that human cathepsin L gene (*CTSL*) encodes two different cathepsins, namely cathepsin L1 (CTSL) and cathepsin V (CTSV). Studies demonstrated that human CTSV evolved from CTSL after mammalian and shares 80% of the homologous to mouse Ctsl due to the amino acid sequence and pattern [15]. The expression of plasma CTSV decreased in patients with abdominal aortic aneurysms [16], while increasing in the stenotic aortic valves [17]. In addition, exogenous CTSV may protect cardiomyocytes from angiotensin II-induced hypertrophy [18], while inhibition of Ctsl may attenuate angiotensin II-induced hypertension [19]. Therefore, the role of CTSV in the cardiovascular system is complicated and remains to be further studied. Previous studies reported that Ctsl played an essential role in retinal and choroidal neovascularization, indicating that Ctsl may be involved in EC function [20]. Recent findings suggested that CTSV decreased in the skin tissue of aged patients [21], while the mechanism of Ctsl/CTSV in aging is elusive. Since ECs senescence is the crucial step of the aging process, we speculated that the Ctsl/CTSV may regulate the ECs senescence to lead to aging. 

In the present study, we found Ctsl expression decreased in the artery of aging mice. We used adenovirus to overexpress CTSV in ECs and further confirmed CTSV deficiency induced ECs senescence. Transcriptome analysis showed that endothelial CTSV participate in ECs senescence by regulating ALDH1A2 via activating AKT/ERK1/2-P21 pathway. Therefore, our results reveal that Ctsl/CTSV may emerge as a promising potential target of EC senescence.

## 2. Materials and Methods

### 2.1. Mice

All animal study experiments were carried out conforming to the National Institutes of Health guidelines and were approved by the Animal Care and Utilization Committee of Xiangya Hospital, China. The 18-month-old and 8-week-old C57 BL/6 J mice were purchased from Vita River laboratory. The mice were sacrificed with 1% pentobarbital sodium. Thoracic aortas were isolated and cut into two sections. One fixed in 4% paraformaldehyde for histology and immunohistochemistry (IHC), while the other stored in liquid nitrogen for western blot (WB) analysis.

### 2.2. Cell Culture and Treatments

Primary human vein endothelial cells (HUVECs) were obtained from a science cell research laboratory and cultured with endothelial cell medium (ECM) containing 5% fetal bovine serum (FBS) and 1% endothelial cell growth supplement (ECGS) at 37 °C and 5% CO_2_. P3-P6 HUVECs were used to our experiment and stimulated with hydrogen peroxide (H_2_O_2_, 323381, Sigma-Aldrich, St. Louis, MO, USA) or Doxorubicin (DOX, E2516, Selleck, Houston, TX, USA) to induce senescence in vitro. For adenovirus infection, recombinant CTSV adenovirus (ad-CTSV, Shanghai Genechem Co., Ltd., Shanghai, China) or negative control adenovirus (ad-NC, Shanghai Genechem Co., Ltd., China) was added to 60% confluent cultured HUVECs and removed after 8 h. Subsequently, HUVECs were cultured in complete growth medium for 40 h before being simulated with 100 nM DOX or 400 μM H_2_O_2_ and 1 μM retinoic acid (RA) for another 24 h [22]; For RNA silencing, CTSV siRNAs (si-CTSV, RiboBio Co., Ltd., China), ALDH1A2 siRNAs (si-ALDH1A2, RiboBio Co., Ltd., Guangzhou, China), or negative control siRNAs (si-NC, RiboBio Co Ltd., China) for 24 h before other treatments.

### 2.3. Histology and IHC

The fixed aortic tissues were embedded in paraffin blocks and cut into 4 µm sections. Hematoxylin-eosin staining, Masson’s trichrome stain (G1006, Servicebio, Wuhan, China), and IHC were conducted as previously described [23,24]. Goat anti-CTSL antibody (1:200 dilution, AF1515, R&D Systems, Minneapolis, MN, USA) was used as primary antibody for IHC, secondary antibody (PV-9003, ZSGB-BIO, Beijing, China) and DAB reagents (ZLI-9017, ZSGB-BIO, China) were used subsequently. The medium thickness (MT), the ratio of medium thickness to lumen diameter (MT/LD), the medium area (MA), the ratio of collagen to the medium area (collagen/MA), and the expression level of CTSL were analyzed using Image J.

### 2.4. Senescence-Associated β-Galactosidase (SA-β-gal) Staining

SA-β-gal staining was conducted according to the manufacturer’s instructions (CST9860S, Cell Signaling Technology, Danvers, MA, USA) [25]. First, 1 × 10^6^ HUVECs were seeded in 6-well plates, cells were rinsed once in PBS after treatments, and fixed in fixative solution for 15 min. After rinsing twice in PBS, fresh SA-β-gal staining solution was added and the plate was placed at 37 °C for 16 h. For each sample, total cells and SA-β gal positive cells were counted in three random visual fields using Image J. The percentage of SA-β gal positive cells was calculated as the ratio of SA-β-gal positive cells to total number of HUVECs.

### 2.5. Wound Healing Assay

First, 5 × 10^5^ cells were seeded in 12-well plates. After treatments, clear wounds were created using the end of 200 μL pipette tips. Images of different times were captured using a phase-contrast microscope (Eclipse, Nikon, Tokyo, Japan).

### 2.6. 5-Ethynyl-20-Deoxyuridine (EdU) Assay

First, 2.5 × 10^5^ cells were seeded in 24-well plates. Meilun EdU Cell Proliferation Kit with Alexa Fluor 555 (MA0425, Meilunbio, Dalian, China) cell was used for EdU assay. A fluorescence microscope was used to obtain images (Eclipse, Nikon, Japan) and Image J was used for quantitation.

### 2.7. Western Blot

Cells in 60 mm plates and aortic tissues were collected in RIPA buffer (P0013B, Beyotime Institute of Biotechnology, Haimen, China) with protease and phosphatase inhibitor cocktail (P1045, Beyotime Institute of Biotechnology, China). Then, 30–40 μg protein was loaded on 12% gel, separated by electrophoresis, and transferred to polyvinylidene fluoride membranes (IPVH00010, Millipore, Darmstadt, Germany). The membranes were incubated with 5% non-fat milk for an hour, primary antibodies overnight at 4 °C, and relevant horseradish peroxidase (HRP)-conjugated secondary antibodies (1:5000 dilution, ZB-2305/ZB-2301, ZSGB-BIO, China) for an hour. The blot bonds were visualized by Immobilon Crescendo Western HRP substrate (WBLUR0500, Millipore, Burlington, MA, USA) and a gel documentation system (Bio-Rad, Hercules, CA, USA). Primary antibodies used in this experiment were as follows: Mouse anti-CTSV antibody (1:1000 dilution, MAB10801, R&D systems, USA), rabbit anti-ALDH1A2 antibody (1:1000 dilution, ab156019, abcam, Cambridge UK), rabbit anti-p21 antibody (1:1000 dilution, ab109520, abcam, UK), rabbit anti-p16 antibody (1:1000 dilution, ab51243, abcam, UK), rabbit anti-p53 antibody (1:1000 dilution, 21891-1-AP, Proteintech, Wuhan, China), rabbit anti-ERK1/2 antibody (1:1000 dilution, CST4695S, Cell Signaling Technology, USA), rabbit anti-phospho-ERK1/2 (Thr202/Tyr204) antibody (1:1000 dilution, CST4370S, Cell Signaling Technology, USA), rabbit anti-AKT antibody (1:1000 dilution, sab4500797, Sigma-Aldrich, USA), rabbit anti-AKT (phospho S473) antibody (1:1000 dilution, ab18206, abcam, UK), goat anti-CTSL antiboty (1:1000 dilution, AF1515, R&D systems, USA), mouse anti-α-tubulin antibody (1:5000 dilution, 66031-1-Ig, Proteintech, China), and rabbit anti-GAPDH antibody (1:10,000 dilution, AP0063, Bioworld, Nanjing, China).

Original Western blots used in this manuscript were uploaded as “Appendix A”.

### 2.8. Quantitative Real-Time PCR (qRT-PCR)

First, 1 × 10^6^ HUVECs were seeded in 6-well plates, total RNA was isolated from cells using AG RNAex Pro Reagent (AG21102, Accurate Biology, Kunming, China) with instructions of the manufacturer. Nano Drop 2100 (Thermo Fisher Scientific, Waltham, MA, USA) was used to qualify and quantify total RNA. cDNA was obtained using an Evo M-MLV Mix Kit with gDNA Clean for qPCR (AG11728, Accurate Biology, China) and the mRNA expression was quantified by qRT-PCR using SYBR Green Premix Pro Taq HS qPCR (Rox Plus) Kit (AG11718, Accurate Biology, China). Relative expression levels were obtained using the 2−ΔΔCt method and normalized by β-actin. The primers involved in this study were synthesized by Sangon Biotech (Shanghai, China) and are shown in Appendix A.

### 2.9. RNA Sequencing and Bioinformatic Analysis

Total RNA was isolated using Trizol (Invitrogen, Waltham, MA, USA) and purified using Oligo(dT)-attached magnetic beads. A-tailing mix and RNA index adapters were added after RNA fragmentation and cDNA generation. Agilent Technologies 2100 bioanalyzer was used for quality control and a library was constructed subsequently. BGIseq500 platform (BGI-Shenzhen, Shenzhen, China) was used to sequence the library. The sequencing data were filtered with SOAPnuke (v1.5.2) [26]. Differentially expressed genes were recognized using the DESeq2 (v1.4.5) with *Q* value < 0.05 [27]. Heatmap was drawn by pheatmap (v1.0.8). 

### 2.10. Intracellular Retinoic Acid (RA) Analysis

First, 5 × 10^5^ cells were seeded in 12-well plates. After treatments, cells were lysed with ultrasonicator on ice and the supernatants were collected. Concentration of retinoic acid was measured using a human retinoic acid ELISA kit (CSB-E16712h, CUSABIO, Wuhan, China) according to the manufacture’s description [28].

### 2.11. Statistical Analyses

Data were expressed as mean ± SEM. If the data were normally distributed, *t*-test or one-way ANOVA with Tukey Kramer test was used; otherwise, non-parametric test was applied. Data analyses were performed by ImageJ 1.51j8, SPSS version 23 or Graph Prism 6. *p* value < 0.05 was considered statistically significant.

## 3. Results

### 3.1. Ctsl Expression Is Decreased in Senescent Aorta

To investigate the relationship between Ctsl and vascular aging, we separated the thoracic aortas of 18-month-old and 8-week-old mice. A significant increase of MT, MT/LD, MA, and collagen/MA was observed in the aortas of aged mice by histological analyses (Figure 1A–C), indicating the presence of vascular remodeling. The expression of Ctsl in young and senescent aortas was analyzed by IHC and WB subsequently. As shown in Figure 1D,E, Ctsl significantly decreased in senescent aortas, with increased expression of p21. Therefore, we may speculate that Ctsl participates in the procedure of the vascular aging.

### 3.2. CTSV Expression Is Inhibited in H_2_O_2_-Induced Cellular Senescence

Since ECs senescence has been recognized as a primary process of vascular aging, we explored the connections between CTSV and endothelial cell senescence in vitro. H_2_O_2_ was widely used as stimulation of inducing cell senescence; therefore, in our study, HUVECs were treated with different dose of H_2_O_2_ (0 μM, 50 μM, 100 μM, 200 μM, and 400 μM) for 24 h to induce cellular senescence. With the raised concentration of H_2_O_2_ increased, SA-β-gal positive cells accordingly increased, as well as the expression of P21 and P16, indicating the presence of senescence. Meanwhile, CTSV significantly decreased after treated with 400 μM H_2_O_2_ (Figure 2A,C and Appendix A), thus the concentration of 400 μM H_2_O_2_ was adapted for further study. However, the expression of P53 remained unchanged at both mRNA and protein level, indicating that P53 was not involved in H_2_O_2_-induced cellular senescence, which was consistent with previous studies [29]. These findings demonstrated that CTSV decreases with cellular senescence in vitro.

### 3.3. Over-Expression of CTSV Reduces Cellular Senescence

To further investigate the effects of CTSV on cellular senescence, adenovirus was applied to enhance the expression of CTSV in HUVECs before treating with H_2_O_2_. It is shown in Figure 3D,E that CTSV significantly increased after transfection with ad-CTSV. Meanwhile, SA-β-gal positive cells decreased after over-expression of CTSV (Figure 3A). Moreover, EdU assay and wound healing assay were conducted to determine ECs proliferation and migration were influenced in this process. As shown in Figure 3B,C, H_2_O_2_ treatment impaired cell proliferation and migration, while over-expression of CTSV reversed the effect of H_2_O_2_. These data indicate that CTSV reduces cellular senescence and enhances cell proliferation and migration in HUVECs. Additionally, the increase of P21 induced by H_2_O_2_ was reduced by ad-CTSV at both mRNA and protein level (Figure 3D,E), while the expression of P53 and P16 was unaffected by ad-CTSV (Appendix A). Therefore, CTSV is supposed to reduce cellular senescence via P21-dependent pathways.

### 3.4. CTSV Regulates ECs Senescence via ALDH1A2

To explore the anti-senescence mechanism of CTSV, RNA sequencing analysis was conducted. Compared with the control group, 3706 upregulated and 3580 downregulated genes were recognized after H_2_O_2_ treatment. As for HUVECs treated with H_2_O_2_ and ad-CTSV, 1 upregulated and 11 downregulated genes were recognized compared with H_2_O_2_ group. The Venn diagram shows that 8 genes which were upregulated after H_2_O_2_ treatment were downregulated after ad-CTSV (Figure 4A). The heatmap of these genes is presented in Figure 4B. 

In addition, the mRNA expression of the recognized 8 genes was confirmed by RT-PCR and 5 of which showed the same tendency as RNA sequencing, including *ALDH1A2*, *IGFBP5*, *NOS3*, *THBD*, and *SEMA3F* (Figure 4C and Appendix A). Since the role of ALDH1A2 and SEMA3F has not been widely studied in cellular senescence [30,31,32], we further analyzed the protein level of ALDH1A2 and SEMA3F by WB. As shown in Figure 4D and Appendix A, the protein expression of SEMA3F was basically unchanged, while ALDH1A2 was significantly increased in H_2_O_2_ group and decreased after over-expression of CTSV. Therefore, ALDH1A2 may be a novel mediator of cellular senescence. It was reported that ALDH1A2 mainly acts as an enzyme that catalyzes the synthesis of retinoic acid (RA) from retinaldehyde, while RA can upregulate the expression of p21 via ERK1/2 and AKT pathways [33,34]. We analyzed the expression and phosphorylation of ERK1/2 and AKT subsequently and found that ad-CTSV reduced the increase of ERK1/2 and AKT phosphorylation induced by H_2_O_2_. 

### 3.5. CTSV Attenuates Cellular Senescence through ALDH1A2-AKT/ERK1/2-p21 Pathway

To verify whether CTSV attenuates cellular senescence by ALDH1A2, exogenous RA was added to HUVECs. RA reversed the anti-senescence effect of CTSV in H_2_O_2_ treated cells, as shown by increased SA-β-gal positive cells and impaired cell proliferation and migration (Figure 5A–C). Additionally, RA did not affect the expression of CTSV and ALDH1A2 but reversed the effect of ad-CTSV on P21, p-ERK1/2, and p-AKT (Figure 5D). 

To further confirm the effect of CTSV in cellular senescence, a doxorubicin (DOX)-induced senescence model was used. Administration of DOX induced cellular senescence and impaired cell proliferation and migration. Simultaneously, it downregulated CTSV expression and upregulated ALDH1A2, RA, p-AKT, p-ERK1/2, and P21. Ad-CTSV reversed the effect of DOX, while further administration of RA canceled the function of ad-CTSV. However, ad-CTSV or RA did not affect the expression of P16 increased by DOX. Moreover, DOX, ad-CTSV, and RA had little influence on the expression of CTSL and P53 (Figure 6, Appendix A).

Furthermore, siRNAs were used to verify the causality of downregulation of CTSV in endothelial senescence. Si-CTSV had a similar effect on ECs as DOX. Additionally, inhibition of ALDH1A2 by siRNAs reversed the effect of si-CTSV and DOX (Figure 7, Appendix A). These results indicate that CTSV attenuates cellular senescence via ALDH1A2-AKT/ERK1/2-P21 pathway.

## 4. Discussion

This study provides new insights into the critical role of Ctsl/CTSV in cellular senescence. The main findings are as follows: 1. The expression of Ctsl/CTSV decreased in senescent aorta and endothelial cells; 2. over-expression of CTSV reduced cellular senescence in vitro; 3. CTSV attenuated cellular senescence by enhancing ALDH1A2 expression and activating AKT/ERK1/2 pathway (Figure 8). We firstly revealed that Ctsl/CTSV was decreased in senescent endothelial cells, which enhanced ALDH1A2 expression and activated AKT/ERK1/2-P21 pathway, and therefore promoted cellular senescence, which may play an important role in vascular aging.

Endothelial senescence is the crucial step of vascular aging leading to the advancement of CVD [7]. To access senescence in vitro, SA-β-gal staining was used in this study, as well as cellular functions including proliferation and migration. Consistent with previous studies [13,24], increased SA-β-gal positive cells and decreased cell proliferation and migration abilities were observed in H_2_O_2_-induced senescent cells. The expression of P53, P16, and P21 has been widely recognized as markers of senescence, which are demonstrated to regulate cellular senescence alone or jointly [35]. p-Coumaric acid is reported to suppress cellular senescence via P53 and P16 pathways [36], while ERK1/2 is involved in H_2_O_2_-induced senescence through P53-independent P21 pathway [29]. In the present study, P21 and P16 were increased in senescent HUVECs, while P53 was not involved in H_2_O_2_-induced cellular senescence. In addition, over-expression of CTSV reduced the expression of P21 but not P16, suggesting that CTSV reduced cellular senescence via P21-dependent pathways. Senescent cells may secrete many factors, such as cytokines, chemokines, metalloproteases and extracellular vesicles, to influence cells around them, which is called SASP [13]. Some factors were commonly found upregulated in different senescent cells, including IL-1β, IL-6, IL-8, ICAM-1, etc. [37]. This study found that IL-1β, IL-6, and ICAM-1 were upregulated in DOX-induced senescent ECs, indicating the existence of SASP. Furthermore, previous studies suggested that the physiological function of ECs was impaired during cellular senescence. Li et al. found that H_2_O_2_ significantly inhibited the migration and proliferation of HUVECs [38]. Similarly, the proliferation and migration ability of HUVECs also decreased in Trimethylamine-N-oxide-induced senescent cells [16]. In this article, impaired cell proliferation and migration were also observed in both H_2_O_2_- and DOX-induced senescent ECs. The secretion of metalloproteases by senescent cells may take response to the disability of cell migration [39], which remains to be further studied.

As a pivotal member of cathepsin family, Ctsl is reported to express in endothelial progenitor cells and plays an essential role in retinal and choroidal neovascularization [20], indicating Ctsl may involve in EC functions, while its role in ECs senescence remains elusive. Ctsl increased in the renal tissue of aged rat as well as the muscle mass of aged mice [40,41], while in the rat brain tissues, the expression and activity of Ctsl decreases with age [42]. In this study, we analyzed the expression of Ctsl in aged thoracic aortas and found that Ctsl obviously decreased in senescent aorta, which was consistent with a previous study [43]. 

Human CTSV evolved from mouse Ctsl, and they have 80% homology [15]. Since CTSV is specifically expressed in human, the role of CTSV in senescence is still unknown. Previous studies showed that CTSV is highly expressed in endothelial cell [44], while its role in endothelial cells and vascular tissues is not clear. In addition, CTSV decreased in the skin tissue of aged patients [21], suggesting that the expression of CTSV may participate in the process of age. Here in our study, we found that CTSV decreased in senescent HUVECs, while over-expression of CTSV reduced cellular senescence, and enhanced cell proliferation and migration. These results indicated that CTSV plays a crucial role in ECs senescence.

Among the 19 members of the aldehyde dehydrogenase super family, aldehyde dehydrogenase 1 family member A2 (ALDH1A2, also known as RALDH2) is one of the most widely distributed members in human tissues [45]. It is reported that ALDH1A2 increases in the hippocampus of senescent female mice [46]. In contrast, Takano et al. demonstrate that ALDH1A2 decreases with age in mesenteric lymph node dendritic cells [47]. In our work, we firstly found that ALDH1A2 was increased in H_2_O_2_-induced senescent endothelial cells by using RNA sequencing and confirmation with RT-PCR and WB, while it decreased after over-expression of CTSV, which suggested the significance of ALDH1A2 in the regulation of cellular senescence by CTSV. As an enzyme, the main function of ALDH1A2 is to catalyze the synthesis of RA from retinaldehyde. Isotretinoin (13-cis-RA), the isoform of RA, has been widely applied in the treatment of severe, recalcitrant acne. However, Bershad et al. noticed that dyslipidemia occurs during isotretinoin therapy for acne, including increased cholesterol and low-density-lipoprotein cholesterol and decreased high-density-lipoprotein cholesterol in plasma [48]. In addition, plasma homocysteine increases significantly after isotretinoin treatment [49]. These changes would increase the risk of CVD if sustained over a long period, indicating the potential unfavorable effects of RA on cardiovascular system. It has been reported that RA decreases in ALDH1A2 knockout endothelial cells, leading to decreased p21 expression, while administrating RA reverses this effect [50]. Furthermore, RA can upregulate p21 to induce cellular senescence via ERK1/2 and AKT pathways in neuroblastoma cells [34]. In the present study, RA was added to CTSV over-expressed senescent ECs. We found that administration of RA significantly activated AKT and ERK1/2, induced the expression of P21, and enhanced ECs senescence. Our results suggested that CTSV may attenuate ECs senescence via ALDH1A2-AKT/ERK1/2-P21 pathway.

As for the limitations, studies on over-expressed Ctsl and genetic knockdown mice should be conducted to further validate our results. Additionally, the regulatory mechanisms of CTSV on ALDH1A2 should be further studied.

## 5. Conclusions

We have demonstrated that decreased Ctsl/CTSV is associated with vascular aging in mice. Moreover, our results suggest that decreased Ctsl/CTSV induces cellular senescence through ALDH1A2-AKT/ERK1/2-P21 pathway. Thus, Ctsl/CTSV may be a potential therapeutic target for EC senescence.

## Figures and Tables

**Figure 1 biology-12-00042-f001:**
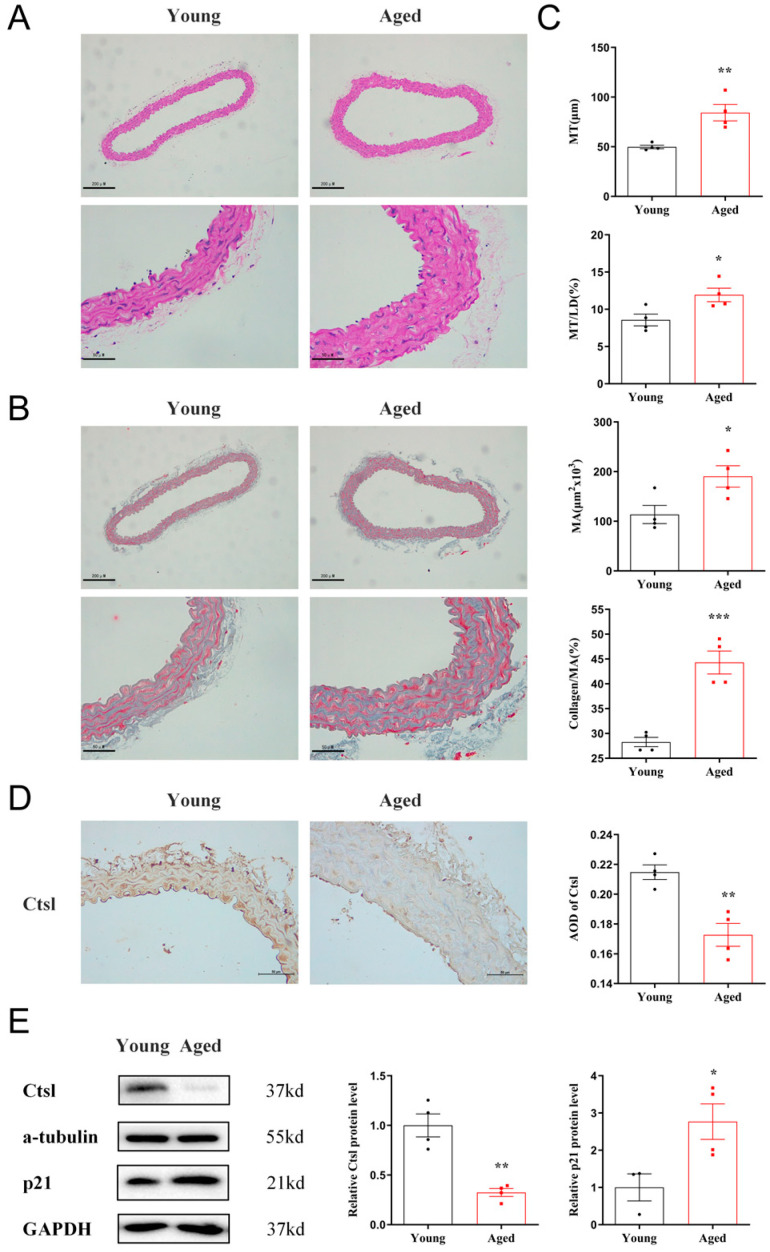
Ctsl expression is decreased in senescent aorta. (**A**) Representative images of hematoxylin-eosin staining in the thoracic aortas of aged (18-month-old) and young (8-week-old) mice (scale bar = 200 μm in top and 50 μm in bottom, respectively). (**B**) Representative images of Masson’s trichrome stain in the thoracic aortas (blue-gray staining for the collagens, scale bar = 200 μm in top and 50 μm in bottom, respectively). (**C**) Quantification of media thickness (MT), the ratio of media thickness to lumen diameter (MT/LD), media area (MA), and the ratio of collagen to media area (collagen/MA) using hematoxylin-eosin staining and Masson’s trichrome stain (*n* = 4). (**D**). Immunohistochemistry of Ctsl in the thoracic aortas of aged and young mice (brown staining for Ctsl, scale bar = 50 μm, *n* = 4). (**E**) Western Blot of Ctsl and p21 in young and senescent aortas (*n* = 3–4). Data were presented as mean ± SEM. *T*-test was used. * *p* < 0.05 vs. young, ** *p* < 0.01 vs. young, *** *p* < 0.001 vs. young.

**Figure 2 biology-12-00042-f002:**
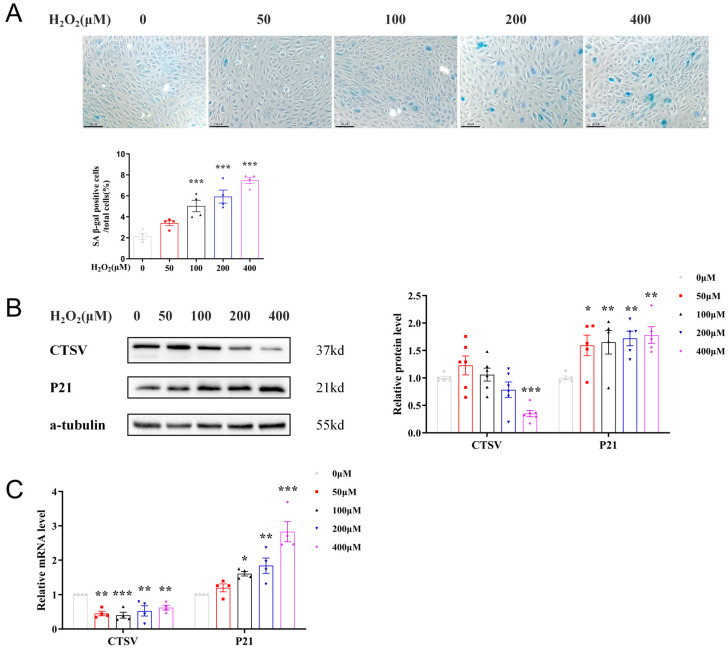
CTSV expression is decreased in H_2_O_2_-induced cellular senescence. (**A**) HUVECs were treated with different dose of H_2_O_2_ (0 μM, 50 μM, 100 μM, 200 μM, and 400 μM) for 24 h, SA-β-gal activity was analyzed (blue staining for the senescent cells, scale bar = 50 μm, *n* = 4). (**B**) Western blot of CTSV and P21 in HUVECs treated with different dose of H_2_O_2_ (*n* = 5–6). (**C**) Relative mRNA level of CTSV and P21 in H_2_O_2_-treated HUVECs (*n* = 4). Data are presented as mean ± SEM. One-way ANOVA test was used. * *p* < 0.05 vs. 0 μM, ** *p* < 0.01 vs. 0 μM, *** *p* < 0.001 vs. 0 μM.

**Figure 3 biology-12-00042-f003:**
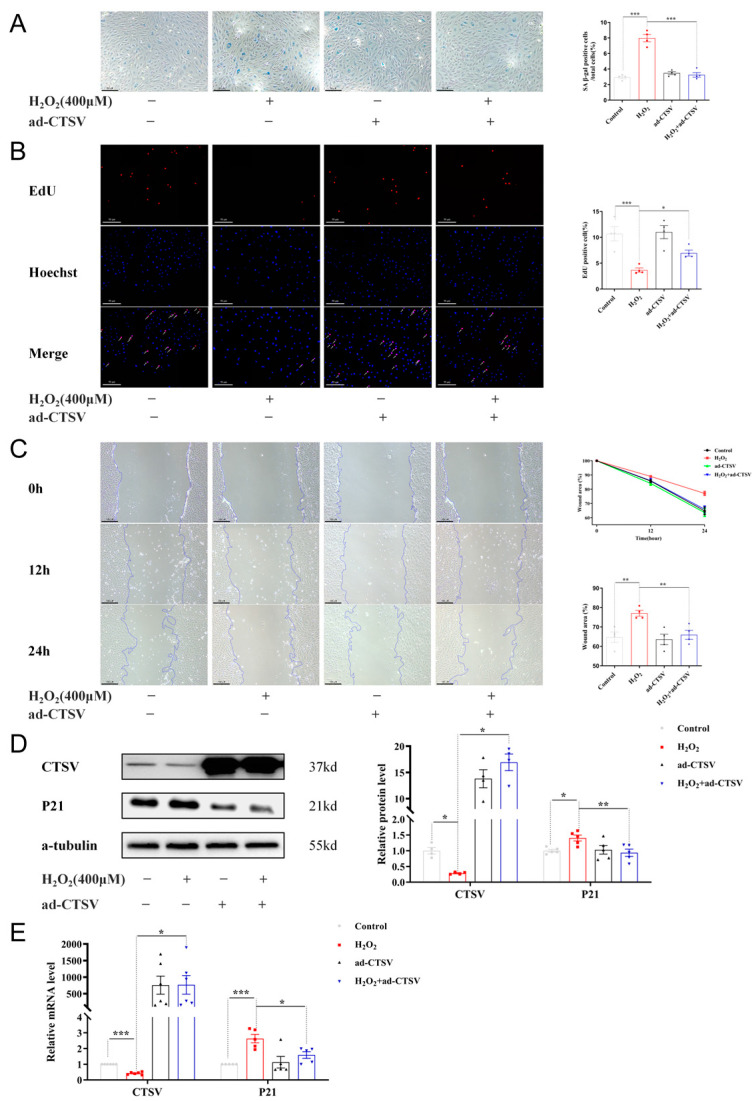
Over-expression of CTSV reduces cellular senescence. (**A**) HUVECs were treated with ad-negative control (NC) or ad-CTSV (MOI = 10) for 8 h and simulated with 400 μM H_2_O_2_ for 24 h after incubated in complete growth medium for 40 h. SA-β-gal activity was analyzed (blue staining for the senescent cells, scale bar = 50 μm, *n* = 4). (**B**) EdU assay of the cell proliferation ability in HUVECs treated with H_2_O_2_ and ad-CTSV (red staining for the EdU, blue staining for Hoechst, scale bar = 50 μm, *n* = 4). (**C**) Representative images of the wound healing assay and the quantification of wound area in HUVECs (scale bar = 100 μm, *n* = 4). (**D**) Western blot of CTSV and P21 in HUVECs treated with H_2_O_2_ and ad-CTSV (*n* = 4–5). (**E**) Relative mRNA level of CTSV and P21 in HUVECs (*n* = 5–6). Data are presented as mean ± SEM. One-way ANOVA test was used. * *p* < 0.05, ** *p* < 0.01, *** *p* < 0.001.

**Figure 4 biology-12-00042-f004:**
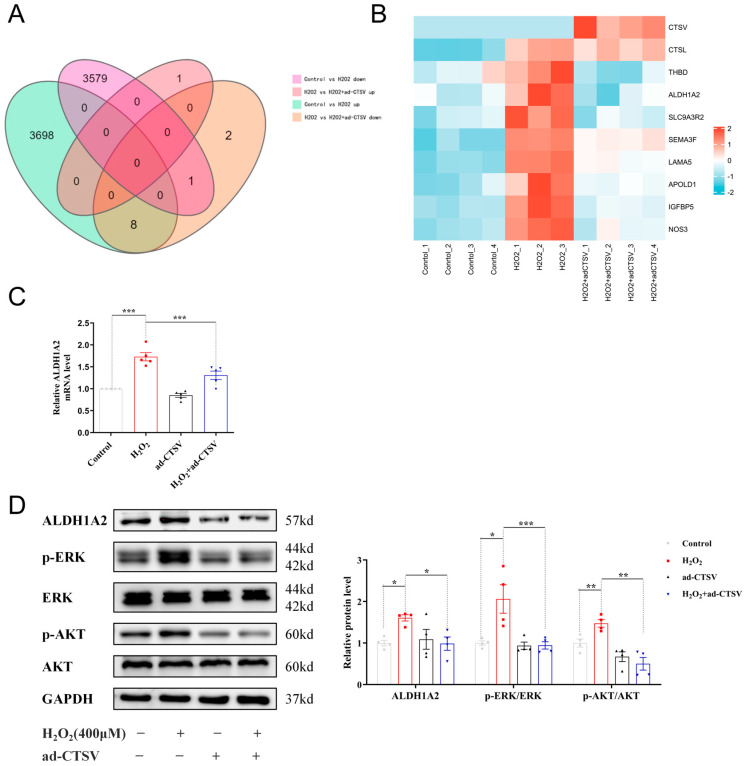
CTSV regulates ECs senescence via ALDH1A2. (**A**) HUVECs were treated with ad-negative control (NC) or ad-CTSV (MOI = 10) for 8 h and simulated with 400 μM H_2_O_2_ for 24 h after incubated in complete growth medium for 40 h. RNA sequencing analysis was conducted and Venn diagram of differentially expressed genes was presented. (**B**) Heatmap of the 8 genes that upregulated after H_2_O_2_ treatment and downregulated after additional ad-CTSV treatment. (**C**) Relative mRNA level of ALDH1A2 in HUVECs treated with H_2_O_2_ and ad-CTSV (*n* = 5). (**D**) Western blot of ALDH1A2, p-ERK1/2, ERK1/2, p-AKT, and AKT in HUVECs (*n* = 4). Data are presented as mean ± SEM. One-way ANOVA test was used. * *p* < 0.05, ** *p* < 0.01, *** *p* < 0.001.

**Figure 5 biology-12-00042-f005:**
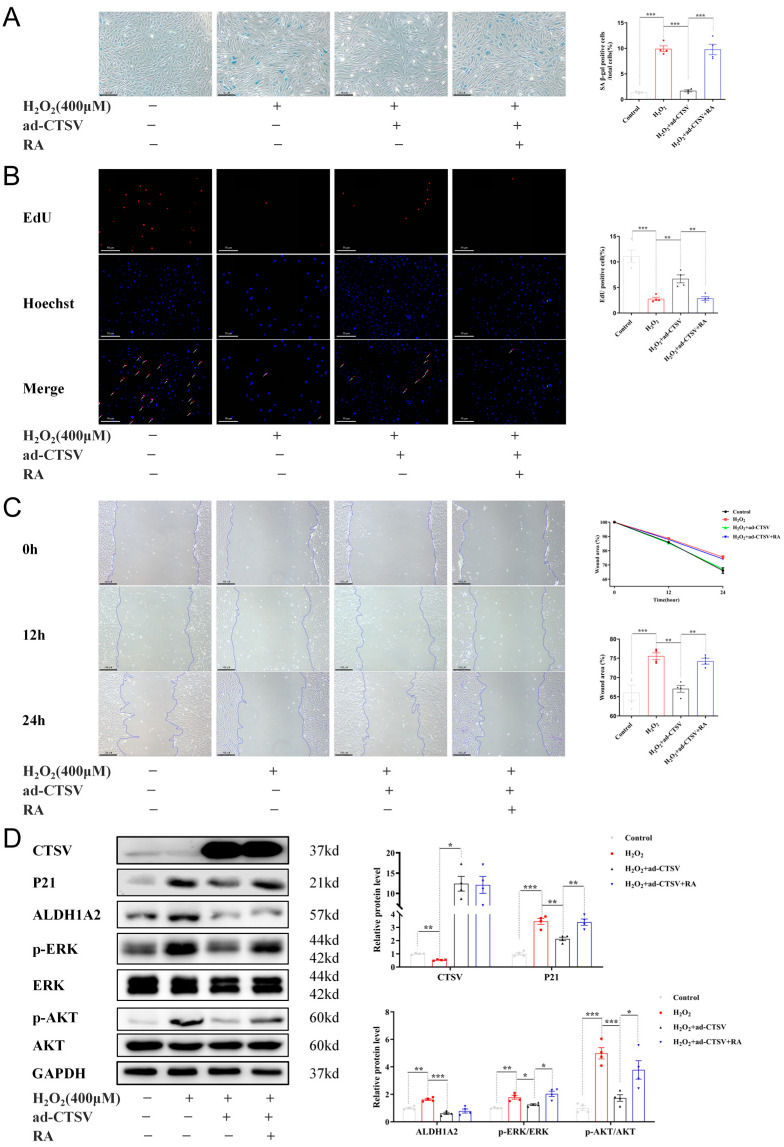
CTSV regulates cellular senescence through ALDH1A2-AKT/ERK1/2-p21 pathway. (**A**) HUVECs were treated with ad-negative control (NC) or ad-CTSV (MOI = 10) for 8 h and simulated with 400 μM H_2_O_2_ and 1 μM RA for 24 h after incubated in complete growth medium for 40 h. SA-β-gal activity was analyzed (blue staining for the senescent cells, scale bar = 50 μm, *n* = 4). (**B**) EdU assay of the cell proliferation ability in HUVECs treated with H_2_O_2_, RA, and ad-CTSV (red staining for the EdU, blue staining for Hoechst, scale bar = 50 μm, *n* = 4). (**C**) Representative image of the wound healing assay and the quantification of wound area in HUVECs (scale bar = 100 μm, *n* = 4). (**D**) Western blot of CTSV, P21, ALDH1A2, p-ERK1/2, ERK1/2, p-AKT, and AKT in HUVECs treated with H_2_O_2_, RA, and ad-CTSV (*n* = 4). Data are presented as mean ± SEM. One-way ANOVA test was used. * *p* < 0.05, ** *p* < 0.01, *** *p* < 0.001.

**Figure 6 biology-12-00042-f006:**
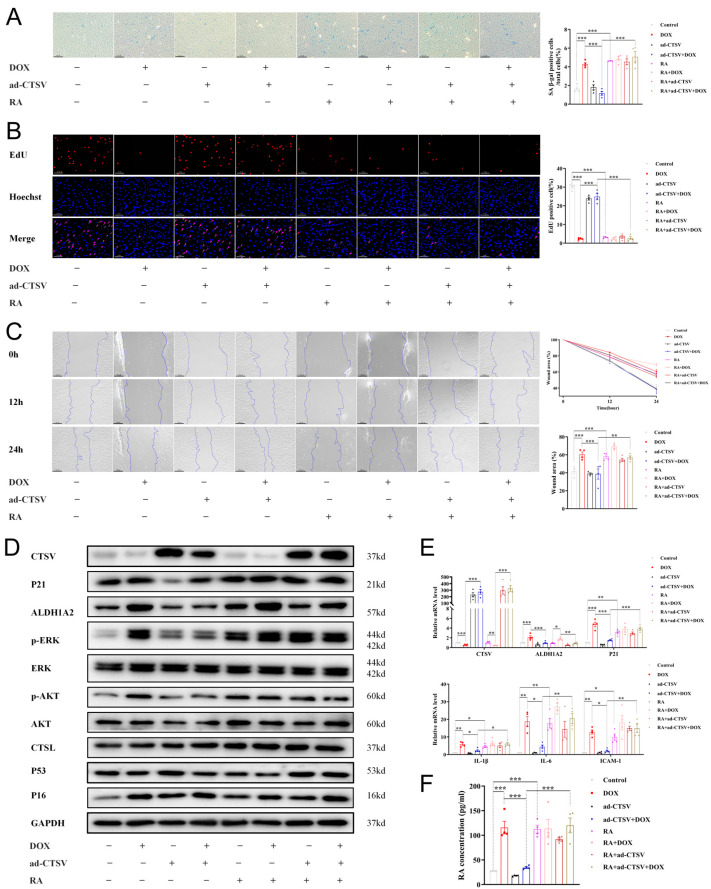
CTSV attenuates DOX-induced cellular senescence. (**A**). HUVECs were treated with ad-negative control (NC) or ad-CTSV (MOI = 10) for 8 h and simulated with 100 nM doxorubicin (DOX) and 1 μM RA for 24 h after incubated in complete growth medium for 40 h. SA-β-gal activity was analyzed (blue staining for the senescent cells, scale bar = 50 μm, *n* = 4). (**B**) EdU assay of the cell proliferation ability in HUVECs treated with DOX, RA and ad-CTSV (red staining for the EdU, blue staining for Hoechst, scale bar = 25 μm, *n* = 4). (**C**) Representative image of the wound healing assay and the quantification of wound area in HUVECs (scale bar = 100 μm, *n* = 4). (**D**) Western blot of CTSV, ALDH1A2, P21, p-ERK1/2, ERK1/2, p-AKT, AKT, CTSL, P53, and P16 in HUVECs treated with DOX, RA and ad-CTSV (*n* = 4). (**E**) Relative mRNA level of CTSV, ALDH1A2, P21, IL-1β, IL-6, and ICAM-1 (*n* = 4). (**F**) RA concentration in HUVECs treated with DOX, RA, and ad-CTSV (*n* = 4). Data are presented as mean ± SEM. One-way ANOVA test was used. * *p* < 0.05, ** *p* < 0.01, *** *p* < 0.001.

**Figure 7 biology-12-00042-f007:**
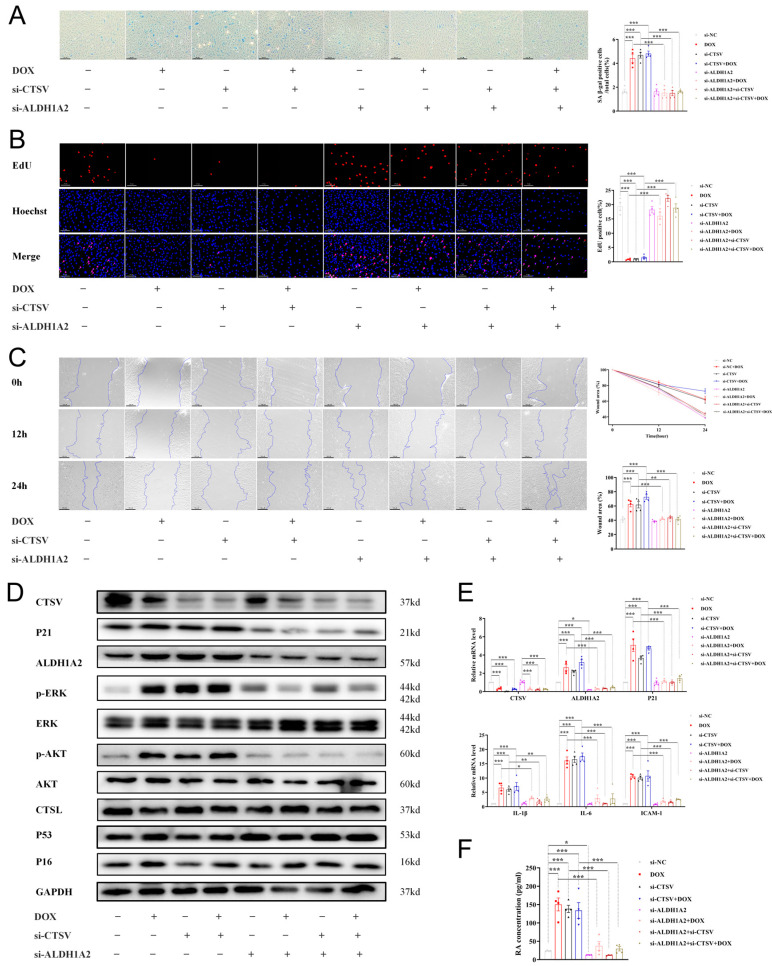
Inhibition of CTSV induces cellular senescence. (**A**) HUVECs were treated with si-negative control (NC), si-CTSV, si-ALDH1A2 or si-CTSV + si-ALDH1A2 for 24 h and simulated with 100 nM doxorubicin (DOX) for 24 h after incubated in complete growth medium for 24 h. SA-β-gal activity was analyzed (blue staining for the senescent cells, scale bar = 50 μm, *n* = 4). (**B**) EdU assay of the cell proliferation ability in HUVECs treated with DOX, si-CTSV and si-ALDH1A2 (red staining for the EdU, blue staining for Hoechst, scale bar = 25 μm, *n* = 4). (**C**) Representative image of the wound healing assay and the quantification of wound area in HUVECs (scale bar = 100 μm, *n* = 4). (**D**) Western blot of CTSV, ALDH1A2, P21, p-ERK1/2, ERK1/2, p-AKT, AKT, CTSL, P53 and P16 in HUVECs treated with DOX and siRNAs (*n* = 4). (**E**) Relative mRNA level of CTSV, ALDH1A2, P21, IL-1β, IL-6 and ICAM-1 (*n* = 4). (**F**) RA concentration in HUVECs treated with DOX and siRNAs (*n* = 4). Data are presented as mean ± SEM. One-way ANOVA test was used. * *p* < 0.05, ** *p* < 0.01, *** *p* < 0.001.

**Figure 8 biology-12-00042-f008:**
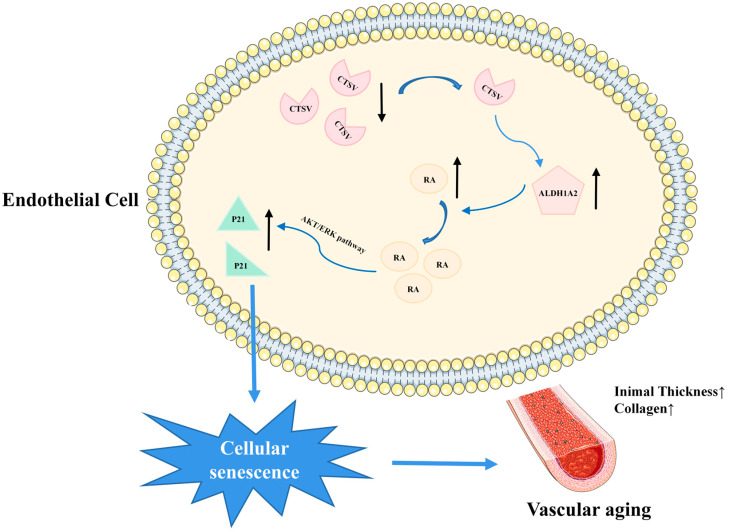
Schematic depiction of the mechanism of CTSV in endothelial cell senescence. CTSV decreased in senescent endothelial cells, leading to increased ALDH1A2, activated AKT/ERK1/2, enhanced P21 expression, and therefore promoted cellular senescence and vascular aging.

## Data Availability

All data generated or analyzed during this study are included in this published article (and its Appendix A files).

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
