# Peer review of "Critical Role of Cathepsin L/V in Regulating Endothelial Cell Senescence"

_biology, 2022, doi:10.3390/biology12010042_

Round 1
Reviewer 1 Report
I want to congratulate the authors on this nice work which suggests that Cathepsin V interferes with the process of senescence in endothelial cells. It will be interesting to find out the vascular phenotype of mice with a Ctsl overexpression or genetic knockdown.
The methods were adequately described and the results were clearly presented. I have one remark about the introduction. I suggest including more information on:
- The different mechanisms to induce in vitro cell senescence. Thus the different inducers (i.e DNA damage, disrupting heterochromatin,...) As well as give more information on the hallmarks of the senescence phenotype (i.E. SA-b-GAL, P53 upregulation,...).
- Give more information on CTSV in the cardiovascular system. For example, cathepsins are also potent elastases and are associated with abdominal aortic aneurysms. This is important to keep in mind when you will work with mice having an overexpression of Ctsl.
Author Response
Thanks for your suggestions. We have included more information about the mechanisms and inducers as well as hallmarks of senescence (page 2). In addition, the role of CTSV in the cardiovascular system has also been introduced in the introduction section (page 2-3).
Reviewer 2 Report
In the paper of Li et al., the role of Cathepsin L/V in regulating endothelial cell senescence that leads to aging by regulating ALDH1A2 and activating AKT/ ERK1/2 -P21 pathway has been studied.
Interestingly, the expression of Ctsl and CTSV was altered. Similarly, ERK1/2 and AKT phosphorylation induced by H2O2-induced senescence were affected. Several interesting data support the mechanism of CTSV in endothelial cell senescence found by the authors.
I would like to kindly ask you the authors to discuss more on the involvement of cell migration and on other possible processes involved in senescence in vitro.
Author Response
Thanks for your suggestions. We have discussed the processes involved in senescence including cell migration and proliferation in the discussion section (page 14-15).